# A Review of Inactivated COVID-19 Vaccine Development in China: Focusing on Safety and Efficacy in Special Populations

**DOI:** 10.3390/vaccines11061045

**Published:** 2023-05-31

**Authors:** Lidan Hu, Jingmiao Sun, Yan Wang, Danny Tan, Zhongkai Cao, Langping Gao, Yuelin Guan, Xiuwei Jia, Jianhua Mao

**Affiliations:** Department of Nephrology, The Children’s Hospital, Zhejiang University School of Medicine, National Clinical Research Center for Child Health, Hangzhou 310030, China; hulidan@zju.edu.cn (L.H.); 22118498@zju.edu.cn (J.S.); wangyan-@zju.edu.cn (Y.W.); tand@dulwich.org.uk (D.T.); rayoflightfever@163.com (Z.C.); 12118249@zju.edu.cn (L.G.); 21807008@zju.edu.cn (Y.G.); jiaqiqi640@163.com (X.J.)

**Keywords:** COVID-19, inactivated vaccines, development process, safety and efficacy, special populations

## Abstract

The coronavirus disease 2019 (COVID-19) pandemic, caused by the severe acute respiratory syndrome coronavirus 2 (SARS-CoV-2), has been widespread globally, and vaccination is critical for preventing further spread or resurgence of the outbreak. Inactivated vaccines made from whole inactivated SARS-CoV-2 virus particles generated in Vero cells are currently the most widely used COVID-19 vaccines, with China being the largest producer of inactivated vaccines. As a result, the focus of this review is on inactivated vaccines, with a multidimensional analysis of the development process, platforms, safety, and efficacy in special populations. Overall, inactivated vaccines are a safe option, and we hope that the review will serve as a foundation for further development of COVID-19 vaccines, thus strengthening the defense against the pandemic caused by SARS-CoV-2.

## 1. Introduction

The outbreak of COVID-19 at the end of 2019 has posed a significant global threat to human health and caused substantial economic losses worldwide [1,2,3,4]. The virus has rapidly spread to all corners of the globe, affecting over 200 countries [1,5]. As of 20 March 2022, there have been over 469 million infections and more than 6 million deaths globally [6]. To control the virus’s spread, countries and the World Health Organization (WHO) are actively developing vaccines, which are crucial tools for controlling and defeating the pandemic.

COVID-19 vaccines can be classified into five types: inactivated vaccines (e.g., Sinovac vaccine, SinoPharm vaccine), adenovirus vector vaccines (e.g., ChAdOx1 nCoV-19, Ad26.CoV2.S, Covishield, CrAdOxI, nCoV-19), recombinant subunit vaccines, nucleic acid vaccines (e.g., mRNA-1273), and synthetic peptide vaccines [7]. Among them, inactivated vaccines are traditional and widely used vaccines that work by modifying the virus’s protein sites to produce a non-toxic vaccine that can prevent infection [8]. Since the outbreak of the pandemic, investigation and development of inactivated vaccines for COVID-19 have been ongoing.

The development process of an inactivated vaccine involves growing the virus in large quantities, inactivating or killing it using physical or chemical means, and using it to stimulate the immune response [9]. Several inactivated vaccine candidates have been developed and are in various stages of clinical trials. For example, Sinovac and Sinopharm have developed two inactivated vaccines in China that have been granted emergency use authorization and are being distributed. In India, Bharat Biotech’s Covaxin is another inactivated vaccine. Other countries, such as Brazil and Turkey, are also developing their own inactivated vaccines against COVID-19. The WHO has emphasized the importance of inactivated vaccines, especially for low- and middle-income countries with limited vaccine infrastructure.

Inactivated vaccines have a strong history of safeguarding against infectious diseases and present a promising solution for curbing and ultimately ending the COVID-19 pandemic. The review details the development process of inactivated vaccines, including important steps such as virus isolation, preclinical testing, clinical trials, regulatory approval, and production and distribution. This review provides a comprehensive overview of inactivated COVID-19 vaccines, with a specific focus on their safety and efficacy in special populations, such as children, adolescents, the elderly, cancer patients, and HIV-infected individuals. As such, this review includes the clinical trials of inactivated COVID-19 vaccines, with particular attention paid to three inactivated vaccines produced in China. By highlighting the importance of inactivated vaccines and their development process, we can better understand their potential to combat the COVID-19 pandemic.

## 2. The Inactivated Vaccine Development Process

The process of developing inactivated COVID-19 vaccines involves killing the coronavirus using chemical or physical means, while preserving its antigenic structure. This process renders the virus non-infectious, but still capable of stimulating the body’s immune response and preventing infection [10]. It involves several key steps, such as virus isolation and propagation, inactivation, purification, formulation, preclinical testing, clinical trials, regulatory approval, and finally, production and distribution (Figure 1).

### 2.1. Virus Isolation and Propagation

To produce vaccines, it is necessary to isolate the virus from either a patient or a sample and then propagate it in a laboratory setting until there is a sufficient amount for vaccine production. Inactivated vaccines by Sinovac have utilized the CN2 virus strain, while Sinopharm’s inactivated vaccines have used the HB02 virus strain [11,12]. The process of virus propagation has evolved over time, with animal inoculation previously used, followed by avian embryo cultivation and, presently, cell culture. Both Sinovac and the Beijing Institute of Biological Products employ Vero cells as passage cells in their vaccine production processes.

### 2.2. Inactivation

Viruses can be inactivated using chemical or physical methods such as formaldehyde, beta-propiolactone, or ultraviolet light. This step ensures that the virus does not cause disease but retains its antigenicity [9].

### 2.3. Purification

Purification follows, using methods such as centrifugation and filtration to remove any impurities that may cause adverse reactions in the body, including other cellular components [13]. This ultimately results in obtaining the viral antigen.

### 2.4. Formulation

Once the viral antigen has been obtained, it is mixed with adjuvants and stabilizers to create a vaccine formulation that is safe for human administration. Adjuvants play a crucial role in enhancing the immune response to inactivated vaccines. Adjuvants are substances that are added to vaccines to stimulate the immune system, allowing for a stronger and longer-lasting immune response to the vaccine [14]. There are several different types of adjuvants that can be used in vaccines, including aluminum salts, oil-in-water emulsions, and toll-like receptor agonists [15]. For example, one commonly used adjuvant in inactivated vaccines is aluminum hydroxide. This adjuvant enhances the immune response by promoting the uptake of the vaccine by immune cells and activating the innate immune system [16]. In addition, oil-in-water emulsions, such as MF59, have been shown to increase the production of antibodies and activate immune cells called T cells, leading to a stronger and longer-lasting immune response [17]. 

Adjuvants also play an important role in vaccine development for certain populations that may have a weaker immune response, such as elderly individuals or immunocompromised individuals. Some adjuvants activate pattern recognition receptors on immune cells, which can trigger the release of cytokines and chemokines that activate other immune cells [18]. Other adjuvants may activate antigen-presenting cells, such as dendritic cells, to better process and present the vaccine antigen to the immune system [19]. This can lead to the activation of T cells, which are important for generating long-term immunity. Overall, the use of adjuvants in inactivated vaccines can increase the potency and duration of the immune response, improving the vaccine’s efficacy.

### 2.5. Preclinical Testing

To ensure the safety and efficacy of the inactivated vaccines, their immunogenicity and potential adverse effects are first evaluated in small animal models such as mice, rats, and guinea pigs [20,21]. There are several biological indicators that can be used to evaluate the immunogenicity and safety of vaccine candidates. These include the following: antibody response: the presence and level of specific antibodies in the blood can indicate the immune response to a vaccine; cellular response: the activation of T cells and other immune cells in response to a vaccine can indicate its efficacy [22,23]; viral shedding: the presence and amount of viral particles shed from the body after vaccination can indicate the vaccine’s ability to prevent viral replication and transmission [24]; side effects: monitoring adverse reactions to the vaccine can provide information on its safety and tolerability; clinical efficacy: the vaccine’s ability to prevent infection or reduce the severity of disease in clinical trials can also serve as a measure of its immunogenicity and safety.

Overall, a combination of these biological indicators can provide a comprehensive evaluation of the immunogenicity and safety of vaccine candidates [25]. In addition, the protective effect of the vaccine against COVID-19 is assessed in susceptible animal models such as rhesus monkeys and ferrets [20,21]. These measures are taken to guarantee the safety and efficacy of the vaccine.

### 2.6. Clinical Trials

After preclinical testing, clinical trials are conducted to evaluate the safety and efficacy of the vaccine in humans. These trials consist of several stages, including phase 1, phase 2, and phase 3 trials, where the vaccine’s ability to induce protective antibodies against SARS-CoV-2 is assessed [26,27]. In phase 1, the vaccine is tested for safety and dosage, while phase 2 evaluates its immunogenicity and optimal dosage. In phase 3, the vaccine is tested on a larger population to evaluate its efficacy and safety in a real-world setting [28]. Clinical trials are critical in ensuring that the vaccine is both safe and effective before it is approved for use in the general population.

### 2.7. Regulatory Approval

Once a vaccine has successfully completed the clinical trial phases and demonstrated safety and efficacy, it undergoes a rigorous regulatory approval process before it can be authorized for public use. Regulatory approval involves thorough evaluation and assessment by regulatory agencies, such as the US Food and Drug Administration (FDA) in the United States or the European Medicines Agency (EMA) in Europe [29].

During the regulatory approval process, the vaccine’s data from preclinical and clinical studies, including its safety profile, efficacy results, manufacturing quality, and proposed labeling, are carefully reviewed by regulatory experts. They assess whether the vaccine meets the established regulatory standards and guidelines for safety, efficacy, and quality. Regulatory agencies also consider the vaccine’s risk–benefit profile, taking into account the severity of the targeted disease, the availability of alternative preventive measures, and the potential public health impact [30]. Additionally, the manufacturing facilities and processes are inspected to ensure compliance with Good Manufacturing Practice (GMP) to guarantee consistent quality and safety of the vaccine [31]. Based on this comprehensive evaluation, regulatory agencies make a determination on whether to grant regulatory approval or authorization for the vaccine. The specific regulatory requirements and processes may vary between countries or regions, but the overall goal is to safeguard public health by ensuring that vaccines meet the necessary standards for safety, efficacy, and quality before they are made available to the public.

### 2.8. Production and Distribution

Once approved by regulators, the vaccines are mass-produced and distributed to the public, and high-risk groups such as medical workers and the elderly are often prioritized. In conclusion, developing a COVID-19 vaccine poses several challenges, including the rapidly evolving nature of the virus and the need for a vaccine that is effective against multiple strains of the virus, and vaccine safety and efficacy must be ensured for all populations, including those with underlying health conditions and those with impaired immune system [29,32]. However, the global effort to exploit and allocate a safe and effective vaccine has been a top priority in the fight against the COVID-19 pandemic [33].

## 3. Progress in Inactivated COVID-19 Vaccine Research in China

A significant role can be played by inactivated vaccines in the SARS-CoV-2 vaccine landscape due to their ease of manufacture, scalability, and relatively low cost [34]. China has made substantial progress in the manufacture of inactivated COVID-19 vaccines, with several Chinese pharmaceutical companies having developed their own inactivated vaccines that have completed clinical trials with positive results. Here, we provide a detailed overview of three inactivated vaccines developed in China: Sinovac, Sinopharm, and the Institute of Medical Biology, Chinese Academy of Medical Sciences. We describe the corresponding clinical trials for each vaccine and also mention their efficacy against the Omicron variant.

### 3.1. Sinovac Vaccine

CoronaVac is the world’s most widely used inactivated COVID-19 vaccine and is one of the leading inactivated vaccines used in China [35]. The vaccine consists of two intramuscular injections, administered 28 days apart, and contains an aluminum hydroxide adjuvant. Phase 3 clinical trials have been completed in several countries, including Brazil, Turkey, and Indonesia, with an overall efficacy rate ranging from 50 to 90% depending on the dosing regimen and trial location [27,36,37]. The interim analysis of a phase 3 trial in Turkey showed that the vaccine had an efficacy of 83.5% (95%CI 65.4–92.1%) [27]. However, smaller trials in different countries reported lower efficacy [36,37]. An observational study involving over 10 million Chilean participants showed an estimated efficacy of 70% in preventing COVID-19 and 86–88% efficacy in preventing hospitalization or death [38]. Subsequently, a study from Brazil reported that the vaccine’s efficacy was lower in adults aged 70 or older during the Gamma variant epidemic period (with efficacy of 47%, 56%, and 61% for preventing COVID-19, hospitalization, and death, respectively) [39]. The vaccine has been validated for emergency use in several countries, including China, Brazil, and Indonesia [40].

During the epidemic, the SARS-CoV-2 virus has been continuously mutating, and the global spread of the Omicron variant constitutes a danger to the lives and property of people around the world [41]. While one study has shown that the Sinovac vaccine has lost its neutralizing activity against Omicron, it still contributes to disease control by driving Fc effector functions through vaccine-induced spike protein-specific antibodies [42]. However, another study showed an increased risk of Omicron infection among people aged 18 to 59 who had been vaccinated once, twice, or three times. The authors caution that these results may be due to the limited sample size in the study and the viral nature of Omicron BA.2 [43].

### 3.2. Sinopharm Vaccine

Another inactivated COVID-19 vaccine developed by Sinopharm has also completed phase 3 clinical trials in several countries, including the UAE, Bahrain, and Egypt. This vaccine is an inactivated whole-virus vaccine containing an aluminum hydroxide adjuvant, and it is administered as two doses via intramuscular injection with a 28-day interval. The vaccine’s efficacy rate varies depending on the trial location and ranges from 78 to 86%. A phase 3 efficacy trial showed that the vaccine had an estimated efficacy of 78% (95%CI 65–86%) compared to the placebo group that only received an aluminum hydroxide adjuvant [44]. Only two severe cases were observed, both of which were in the placebo group. The incidence of systemic and injection site reactions, such as pain (20–27%), headache (13%), and fatigue (11%), was similar in both groups. The vaccine has been approved for emergency use in several countries, including China, the UAE, Bahrain, and Egypt.

Recent research indicates that high-dose and persistent two-dose and three-dose inactivated vaccines have limited efficacy in preventing severe/critical illness and death caused by Omicron infection across all age groups [45]. However, a study has demonstrated that both Sinovac’s CoronaVac and Sinopharm’s BBIBP CorV vaccines offer protection against pneumonia and severe illness caused by Omicron, although they do not prevent any symptomatic illness. Moreover, there was no statistically significant difference in efficacy between the two vaccines [46].

### 3.3. Institute of Medical Biology Vaccine

Another inactivated vaccine developed by the Institute of Medical Biology, Chinese Academy of Medical Sciences, has completed phase 2 clinical trials in Yunnan, China. The results show that the vaccine can trigger effective antibody responses, with the high-dose group receiving 0 and 14 procedures showing higher serum conversion rates and GMTs of anti-S and anti-N antibodies [47].

### 3.4. Other Vaccines

Besides the vaccines mentioned earlier, numerous other Chinese pharmaceutical companies, including Clover Biopharmaceuticals and Walvax Biotechnology, are also developing their own inactivated COVID-19 vaccines [48,49]. These vaccines are currently in different phases of clinical trials and are expected to expand the selection of effective inactivated COVID-19 vaccines in the foreseeable future.

Overall, the development of inactivated vaccines against COVID-19 in China is progressing rapidly, with multiple vaccines showing promising results in clinical trials. These vaccines have been approved for use in several countries and are expected to play a key role in global efforts to contain the COVID-19 pandemic. As the SARS-CoV-2 virus is prone to mutation and new subtypes are emerging, continued research is needed to ensure that inactivated vaccines against COVID-19 continue to be effective in preventing COVID-19 and its severe consequences.

## 4. Safety and Efficacy of Inactivated COVID-19 Vaccines

COVID-19 is a worldwide pandemic caused by severe acute respiratory syndrome coronavirus 2 (SARS-CoV-2) that has resulted in a high mortality and morbidity rate [50]. COVID-19 vaccines are crucial in preventing further morbidity and mortality [51]. Due to their safety and immunogenicity, COVID-19 vaccines are widely manufactured and rolled out. In healthy adults aged 18–59 years, CoronaVac has been shown to be well tolerated with moderate immunogenicity. Furthermore, the majority of adverse reactions were slight, with pain at the injection site being the most common symptom, which aligns with the findings from Sinopharm’s (Beijing, China) study of another inactivated COVID-19 vaccine [52,53].

Several clinical trials have been conducted to evaluate the safety and efficacy of inactivated vaccines, including those developed in China. For example, a study reported that the CoronaVac inactivated vaccine developed by Sinovac Biotech in China was 51% effective in preventing symptomatic COVID-19 infections [53] and 100% effective in preventing severe disease and hospitalization in a large-scale clinical trial in Brazil [54]. Similarly, a study reported that the BBIBP-CorV inactivated vaccine developed by the Beijing Institute of Biological Products in China was 78% effective in preventing symptomatic COVID-19 infections and 100% effective in preventing severe disease and hospitalization in a phase 3 trial in the United Arab Emirates [55,56]. These results provide evidence that inactivated vaccines can effectively reduce the risk of infection and hospitalization due to COVID-19.

In the general population, most COVID-19 vaccines available on the market have demonstrated high efficacy and safety. However, it is equally important to gather safety and efficacy data for specific populations, such as children/adolescents, the elderly, chronically ill individuals, and cancer patients, to ensure comprehensive coverage of COVID-19 vaccines. Some studies have shown heterogeneity in the antibody responses of SARS-CoV-2 vaccine recipients with underlying diseases, including cancer, autoimmune disease, and HIV infection [46,57,58,59,60,61,62,63,64,65,66,67,68,69] (Table 1). Table 1 provides a detailed overview of various COVID-19 vaccines, including their targeted populations, drug names, titles of clinical trials, clinical trial phases, current status, sample sizes, trial durations, immunogenicity data, safety data, and adverse effects. This table serves as a valuable resource for readers to access specific information about different vaccines in a concise and organized manner. Among these inactivated COVID-19 vaccines, CoronaVac, WIBP-CorV, and BBIBP-CorV trigger an immune response and require two doses, while SARS-CoV-2 Vaccine requires three doses.

### 4.1. Children and Adolescents

A randomized, double-blind, controlled, phase 1/2 clinical trial was conducted at the Hebei Provincial Center for Disease Control and Prevention to assess the safety and efficacy of COVID-19 vaccines in healthy children and adolescents aged 3–17 years. Pain at the injection site was the most common adverse reaction, which was slight to moderate in severity and resolved quickly. Neutralizing antibodies were converted into positive antibodies in children and adolescents after two doses at a rate of over 96% [61]. These findings suggest that the inactivated vaccine was safe, well tolerated, and highly effective in young people aged 3–17 years. Furthermore, several studies have investigated the impact of age on the response to COVID-19 vaccines in children and adolescents. Adolescents and older children are more likely to experience adverse reactions to vaccines and demonstrate stronger immune responses than younger children [70]. However, there was no difference in the neutralizing antibody response between adolescents aged 12 to 15 years and those aged 16 and older after vaccination [71]. Currently, China has approved several COVID-19 vaccines, and the safety of inactivated vaccines is reported to be better than that of gene vaccines and adenovirus vector vaccines [70]. Each type of vaccine has its own characteristics, and the vaccines’ safety and efficacy are supported by certain data, thus making it safe to vaccinate against the novel coronavirus.

### 4.2. Old People

A randomized, double-blind, placebo-controlled phase 1/2 clinical study was conducted in 60-year-old healthy adults. After two shots of the vaccine, the seroconversion rate was over 95%. Furthermore, most side effects were mild and transient, with pain at the injection site being the most common symptom [72]. The meta-analysis looked at eight randomized controlled trials (RCTs) of vaccines against COVID-19. The geometric mean viral load (GMT) of the elderly group was significantly higher than that of the placebo group, but there was no significant difference between the elderly group and the young and middle-aged group. This group had a lower event rate than the young adult group [73]. These findings suggest that elderly people can safely and effectively receive inactivated vaccines against COVID-19. Despite a large proportion of elderly individuals having underlying medical conditions, they are a key group that needs protection against COVID-19. Therefore, elderly people with basic diseases can be vaccinated against COVID-19 if their basic diseases have been well controlled through conventional treatment.

### 4.3. Cancer Patients

A study was conducted to test antibodies against SARS-CoV-2 in blood samples from 776 cancer patients and 715 non-tumor volunteers. The serum positive rate was found to be 85.2% in the case group and 97.5% in the control group, with a significant difference between the two groups in terms of the serum positive rate and antibody level [66]. Cancer patients may be immunocompromised for unique or treatment-related reasons, making them more susceptible to COVID-19 infection than the general population. Therefore, the seropositivity rate of cancer patients is relatively lower than that of adults without cancer, and the high seropositivity rate of cancer patients also supports early vaccination. The suppression of humoral and T-cell responses by cancer therapies, such as chemotherapy, is not uncommon in solid tumor patients after vaccination [74]. Camrelizumab has mild side effects in cancer patients, does not increase the serious side effects associated with anti-PD-1, and does not reduce clinical efficacy. Therefore, cancer patients should not stop anti-PD-1 therapy during COVID-19 vaccination [75]. Cancer patients, especially those undergoing aggressive healing, should be vaccinated as a matter of priority according to the National Comprehensive Cancer Network (NCCN) and other oncology societies [76].

### 4.4. HIV-Infected Patients

A systematic review and meta-analysis of 22 studies showed that seroconversion rates in HIV-infected individuals after the first vaccination were comparable to those in healthy individuals. After the second vaccination, the seroconversion rate of the HIV group was slightly lower than that of the control group [77]. The study also showed that neutralizing antibody responses in HIV-infected patients peaked later than those in healthy controls. While the immunoglobulin response peaked at the same time in both groups, the early humoral immune response to a COVID-19 inactivated vaccine in the HIV-infected group was weaker and delayed compared to that in the healthy control group [78]. By multivariate analysis, it was found that the factors that predicted weaker response to an inactivated vaccine in HIV-infected patients were a lower CD4 count and a longer interval after vaccination [68]. In conclusion, inactivated vaccines were well tolerated but less immunogenic in HIV-infected patients. Numerous data show that immunocompromised people are at higher risk of becoming sick and dying from the novel coronavirus. People living with HIV should therefore be included in the vaccination population, and those with advanced HIV disease should be considered a high priority. If the virus is well controlled, it is recommended to vaccinate against COVID-19 as soon as possible without considering the level of CD4 cells.

Apart from the groups mentioned above, other priority population groups that should be considered include pregnant women, immunocompromised individuals, individuals with comorbidities, and infected patients. Women who received two doses of an inactivated vaccine in early or perinatal pregnancy effectively produced neutralizing antibodies and had no increase in the incidence of miscarriage, pregnancy complications, or complications during delivery [79]. There was no difference in the incidence of congenital malformations in neonates compared to controls, nor was there a difference in various indicators such as height and weight, although neonatal jaundice should be noted [80]. No serious adverse events were observed in immunosuppressed populations such as liver transplant patients and patients with autoimmune inflammatory rheumatic diseases during or after inactivated vaccination. Liver transplant patients have a partial immune response to an inactivated vaccine and require a booster dose if better vaccine protection is desired [81]. The seropositivity rate in patients with autoimmune inflammatory rheumatic diseases was not statistically significant compared to controls, and their anti-S1/RBD protein IgG antibody levels were comparable to controls with appropriate immunogenicity [82]. HIV-infected patients were described in detail in the previous text. The inactivated vaccine was well tolerated in patients with comorbidities, and patients with chronic hepatitis B and healthy controls had similar seropositivity rates of antibodies at 1, 2, and 3 months in inducing an effective SARS-CoV-2 antibody response [83]. Antibody responses and memory B-cell responses were significantly lower in patients with type 2 diabetes compared to healthy controls [84]. The frequency of Ab concentrations of anti-RBD IgG and CoV-2 neutralizing Abs and the proportion of RBD-specific memory B cells were lower in elderly TB patients than in healthy controls, with limited immunogenicity [65]. Inactivated vaccines provide limited protection against progression from asymptomatic infection to moderate to mild disease in infected populations and durable protection against the progression of non-severe disease to severe disease caused by Omicron BA. Partial vaccination does not provide effective protection under any circumstances [85].

Regarding the duration of protection, the current understanding is that inactivated vaccines provide protection against COVID-19 for at least six months, but more long-term data are needed to determine the full duration of protection [86]. The booster effects of inactivated vaccines are also being studied, and initial findings suggest that booster doses can increase antibody levels and improve protection against emerging variants [87,88]. Several studies have shown that booster doses of inactivated COVID-19 vaccines can enhance the immunity of individuals against the virus [89]. For example, a study conducted in China on individuals who had received two doses of the inactivated COVID-19 vaccine found that a booster dose administered six months after the second dose resulted in a significant increase in antibody levels against the virus [89,90]. Interchangeability refers to the ability of different vaccines to be used interchangeably or substituted for each other in a vaccination schedule. In the case of inactivated COVID-19 vaccines, there are limited data available on their interchangeability with other COVID-19 vaccines. The World Health Organization (WHO) currently recommends that people who have received one dose of an inactivated COVID-19 vaccine should receive a second dose of the same vaccine, if available [91,92]. However, if the same vaccine is not available, the WHO suggests that a different COVID-19 vaccine may be considered to complete the vaccination schedule, based on expert review and individual assessment of the risks and benefits [93].

In terms of variant mutations, inactivated vaccines have been shown to provide some protection against emerging variants, including the Delta and Omicron variants [94]. There have been several studies conducted to assess the efficacy of inactivated COVID-19 vaccines against emerging variants [90]. A study analyzed the efficacy of the Sinovac inactivated COVID-19 vaccine against the Delta variant. The study found that the vaccine had an efficacy of 83.5% against symptomatic COVID-19 caused by the Delta variant [95]. Another study assessed the immune response of individuals who received the Sinovac or Sinopharm inactivated COVID-19 vaccine against the Delta variant [96]. The study found that both vaccines generated neutralizing antibodies against the Delta variant, although the levels of antibodies were lower compared to the original strain [97]. A preprint study evaluated the efficacy of the CoronaVac inactivated COVID-19 vaccine against the Gamma variant. The study found that the vaccine was effective in preventing COVID-19 caused by the Gamma variant, with an overall efficacy of 50.7% [98]. However, ongoing surveillance and research are needed to determine the efficacy of inactivated vaccines against future variants. As with any vaccine, there are some contraindications for inactivated COVID-19 vaccines, such as known allergies to vaccine components [99,100]. However, the safety profile of inactivated vaccines has been generally favorable, with reported adverse events being mostly mild and transient. The contraindications of inactivated COVID-19 vaccines may vary depending on the specific vaccine product and country regulations.

## 5. Discussion

One of the innovative aspects of this review is its exploration of the safety and efficacy of inactivated vaccines in vulnerable populations, such as children, adolescents, older adults, cancer patients, and HIV-infected individuals. The article provides valuable information for vaccine development efforts and emphasizes the importance of vaccine safety and efficacy in protecting vulnerable populations and mitigating virus transmission. Additionally, the article highlights the potential of inactivated vaccines as a promising solution to combat the COVID-19 pandemic due to their ease of manufacturing, scalability, and relatively low cost. The many other types of COVID vaccines in widespread use include mRNA vaccines (Pfizer-BioNTech and Moderna), viral vector vaccines (AstraZeneca and Johnson & Johnson), and protein subunit vaccines (Novavax). These vaccines work by different mechanisms to stimulate an immune response against the SARS-CoV-2 virus. mRNA vaccines, such as Pfizer-BioNTech and Moderna, use a small piece of genetic material called mRNA to stimulate an immune response to the virus. These vaccines have shown high efficacy rates of around 90–95%, with some protection against variants of the virus [101]. Viral vector vaccines, such as AstraZeneca and Johnson & Johnson, use a harmless virus to deliver a piece of the coronavirus genetic material to cells and stimulate an immune response. These vaccines have also shown high efficacy rates, ranging from 60 to 90%, with some protection against variants of the virus [102]. Protein subunit vaccines, such as Novavax and Sanofi/GSK, use a harmless piece of the coronavirus spike protein to stimulate an immune response. These vaccines have shown efficacy rates ranging from 50 to 90%, with some protection against variants of the virus [103]. Compared to these vaccines, inactivated vaccines have generally shown slightly lower efficacy rates, ranging from 50 to 80%, but still offer significant protection against COVID-19. While the inactivated COVID-19 vaccines have been shown to be safe and effective, the other types of vaccines also have their own advantages, such as potentially higher efficacy rates and easier storage requirements. Ultimately, the choice of vaccine type will depend on factors such as local availability, individual medical history, and vaccine preferences [44].

Overall, this review provides a valuable contribution to the ongoing conversation surrounding COVID-19 vaccine development and distribution. The WHO Strategic Advisory Group of Experts (SAGE) has recommended the use of inactivated vaccines as an effective tool to combat COVID-19 [92,104,105]. According to its interim recommendations, inactivated vaccines should be prioritized for use in populations with a high risk of severe disease or death, as well as in settings with limited access to mRNA vaccines [106]. The SAGE also advises that inactivated vaccines may be used as booster doses, particularly for individuals who have received a primary series of the same vaccine [107]. Overall, the use of inactivated vaccines has been shown to be safe and effective in preventing COVID-19 and reducing its transmission.

One limitation of our article is its exclusive focus on inactivated COVID-19 vaccines developed in China, without a detailed discussion of other types of vaccines. Another is that we exclude the logistics, service delivery, and population acceptance of vaccines, which are also critical effectiveness factors for vaccines. Of course, to reduce infections and the impact of the disease, all operational aspects of vaccination delivery have to be considered for the sake of effective vaccination performances and results. However, we focus on safety and efficacy in this study. Moreover, a comparative analysis of different vaccine types could provide insights into the strengths and weaknesses of each approach, facilitating informed decision making in vaccine development and distribution. Therefore, future research could expand on our review by incorporating a more comprehensive evaluation of different COVID-19 vaccines and their implications for controlling the pandemic.

## 6. Conclusions

Inactivated COVID-19 vaccines have been proven to be safe and effective in preventing COVID-19 infections, based on clinical trial results and real-world data. In our review, we provide the overall development process of inactivated vaccines, with a specific focus on safety and efficacy in special populations. As China has been a major producer and user of this type of vaccine, we focus on China’s experience with inactivated COVID-19 vaccines.

Several inactivated COVID-19 vaccines, including those developed by Sinovac and Sinopharm in China, have successfully completed phase 3 clinical trials with positive results. These trials indicate that inactivated COVID-19 vaccines have an efficacy rate ranging from around 50% to 90%, depending on the vaccine and dosing regimen used [44]. While the efficacy rate may be lower than that of some other COVID-19 vaccines, such as mRNA vaccines, it still provides significant protection against COVID-19. Real-world data from countries such as China, Brazil, and the United Arab Emirates, which have approved the use of inactivated COVID-19 vaccines, have demonstrated their efficacy in reducing COVID-19 infections and hospitalizations.

Given the unique susceptibility and potential challenges in the vaccine response of special populations, we highlight the vaccines’ safety and efficacy in special populations. Moreover, safety and efficacy data for inactivated vaccines among different populations, including children and adolescents, elderly individuals, cancer patients, pregnant women, immunocompromised individuals, individuals with comorbidities, and those with HIV infections, have shown that they are safe and effective. While the fundamental principles of vaccine development and evaluation remain the same for all population groups, it is important to note that the overall safety and efficacy of these vaccines are similar across different population groups. The majority of adverse reactions reported were mild, with injection site pain being the most common symptom. By and large, inactivated COVID-19 vaccines are a safe and effective tool in the global effort to control the COVID-19 pandemic. Moreover, the SAGE recommendations for the use of inactivated COVID-19 vaccines emphasize their efficacy in preventing severe disease, hospitalization, and death caused by COVID-19 [104,105,108]. These vaccines are recommended for individuals within the approved age groups and population categories, taking into account factors such as the local epidemiological situation, vaccine supply, and individual risk profiles.

It is important to note that the specific recommendations for the use of inactivated COVID-19 vaccines may be country-specific, as each country’s regulatory authorities and public health agencies evaluate the available evidence and tailor their recommendations accordingly. Therefore, it is advisable to consult the official guidelines and recommendations of your respective national health authorities or the WHO website for the most up-to-date and country-specific information on the use of inactivated COVID-19 vaccines.

## Figures and Tables

**Figure 1 vaccines-11-01045-f001:**
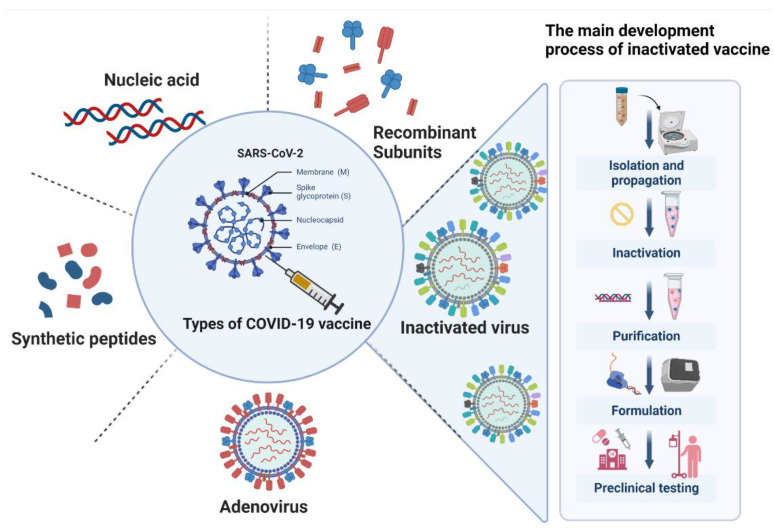
Commonly used types of SARS-CoV-2 vaccines and the detailed process of inactivated vaccine development.

**Table 1 vaccines-11-01045-t001:** Clinical studies concerning COVID-19 vaccines’ safety in special populations identified in clinical trials.

Targeted Populations	Drug Name	Title of Clinical Trial	Clinical Trial/Phase	Status	Sample Size	Clinical Trial Duration	Immunogenicity Data	Safety Data	Adverse Effects	Ref.
Children and adolescents	CoronaVac	Safety and Immunogenicity Study of Inactivated Vaccine for Prevention of COVID-19	NCT04551547/1-2	Active, not recruiting	320	180 days	neutralizing antibodies	well-tolerated	Injection site pain, fever	[61]
Children	WIBP-CorV	A randomized, double-blind, placebo parallel-controlled phase I/II clinical trial for inactivated Novel Coronavirus Pneumonia vaccine (Vero cells)	ChiCTR2000031809/1-2	Ongoing	640	4 months	neutralizing antibodies	well-tolerated	Injection site pain, fever	[62]
Children and adolescents	BBIBP-CorV	A phase I/II clinical trial for inactivated novel coronavirus (2019-CoV) vaccine (Vero cells)	ChiCTR2000032459/1-2	Ongoing	960	6 months	neutralizing antibodies	well-tolerated	Injection site pain, fatigue, headache	[46]
Children and adolescents	CoronaVac	Efficacy, Immunogenicity and Safety of COVID-19 Vaccine, Inactivated in Children and Adolescents	NCT04992260/3	Active, not recruiting	4500	4 months	vaccine elicited robust humoral and cellular immune responses	well-tolerated	Injection site pain	[63]
Older adults	CoronaVac	Safety and Immunogenicity Study of Inactivated Vaccine for Prevention of SARS-CoV-2 Infection (COVID-19)	NCT04383574/1-2	Completed	720	2 months	neutralizing antibodies	well-tolerated	Injection site pain, fever, fatigue	[64]
Old pulmonary tuberculosis patients	SARS-CoV-2 Vaccine	Safety and Immune Response of COVID-19 Vaccination in Patients With Basic Disease (SIM-PBD)	NCT05043246/1	Recruiting	180	4 months	neutralizing antibodies	well-tolerated	Injection site pain	[65]
Cancer patients	COVID-19 Vaccines	Evaluation of the Effect and Side Effect Profile of COVID-19 Vaccine in Cancer Patients	NCT04771559/1-2	Completed	100	9 months	neutralizing antibodies		Injection site pain, fatigue	[66]
HIV-infected patients	inactivated SARS-CoV-2 Vaccine	Immunogenicity and safety of inactivated SARS-CoV-2 vaccine in people living with HIV	ChiCTR2100051956	Completed	300	12 months	cellular immune response	well-tolerated	Injection site pain	[67]
HIV-infected patients	SARS-CoV-2 Vaccine	Safety and Immune Response of COVID-19 Vaccination in Patients With HIV Infection	NCT05043129/2-3	Recruiting	5000	12 months	neutralizing antibodies	well-tolerated	Injection site pain	[68]
HIV-infected patients	Inactivated COVID-19 Vaccine	Immunogenicity and Safety of an Inactivated COVID-19 Vaccine in People With HIV Infected	NCT05075070/4	Recruiting	400	2 months	neutralizing antibody	well-tolerated	Injection site pain	[69]

## Data Availability

Not applicable.

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
