# Peer review of "A Review of Inactivated COVID-19 Vaccine Development in China: Focusing on Safety and Efficacy in Special Populations"

_vaccines, 2023, doi:10.3390/vaccines11061045_

Round 1

Reviewer 1 Report (Previous Reviewer 3)

I appreciate additional information provided as a result of reviewers suggestions, in particular related to some special population groups and the discussion points.

These are important amendments.

The titlle, the summary and the conclusions should clearly indicate the focus of the review on China.

The term effectiveness has not been addressed with scientific evidence .( related to the delivery aspects ) . So, the use of "efficacy" is the appropriate terminology.

The references to the SAGE recommandations should be made explicit and mentionned in the conclusion section.

Author Response

Thank you for your efforts to review our manuscript entitled “A Review of Inactivated COVID-19 Vaccines Development in China: Focusing on Safety and Efficacy in Special Populations” (ID: vaccines-2401664). ”. We highly appreciate your time and comments that help a lot to improve the quality of the manuscript.

 We have addressed the comments from reviewers in a point-by-point way. Please see them in the response letter as below. In the revised manuscript, we have marked the revised texts in red color.

Hopefully, the resived manuscript will meet with the criteria of publication of your well-reputed journal. Thank you in advance.

  1. I appreciate additional information provided as a result of reviewers suggestions, in particular related to some special population groups and the discussion points.

Response:Thank you for your feedback and positive remarks regarding the additional information provided in response to your suggestions. We greatly appreciate your recognition of our efforts to address the reviewers' comments and enhance the value and reliability of the manuscript. We will carefully consider your remaining suggestions and continue to refine and improve the paper accordingly. Thank you for your valuable input and guidance throughout the review process.

  1. These are important amendments.

  1. The title, the summary and the conclusions should clearly indicate the focus of the review on China.

Response:Thank you for your feedback. We appreciate your suggestion regarding the clarity of the title, summary, and conclusions. In response, we have revised the title, summary, and conclusions to clearly indicate the focus of the review on China. The revised sections now explicitly mention the context of the review being centered on the inactivated COVID-19 vaccines in China.

  1. The term effectiveness has not been addressed with scientific evidence.( related to the delivery aspects ) . So, the use of "efficacy" is the appropriate terminology.

Response:Thank you for your suggestion. We have carefully considered your feedback and made the necessary revisions to address the concern regarding the use of the term "effectiveness." In response to your recommendation, we have replaced the term "effectiveness" with "efficacy" throughout the manuscript to ensure appropriate scientific terminology is used.

  1. The references to the SAGE recommandations should be made explicit and mentionned in the conclusion section.

Response:Thank you for your suggestion. We have carefully reviewed the manuscript and made the necessary revisions to address your comment. In the revised version, we have included explicit references to the SAGE recommendations throughout the manuscript (page 11, line 450). Additionally, we have included a specific mention of the SAGE recommendations in the conclusion section to emphasize their importance in guiding global vaccination strategies (page 12, line 494).

Reviewer 2 Report (Previous Reviewer 2)

The authors have revised their manuscript significantly and have responded to my comments.

Author Response

The authors have revised their manuscript significantly and have responded to my comments.

Response:Thank you for acknowledging the significant revisions made to the manuscript and recognizing our efforts in responding to your comments. We have carefully considered your suggestions and feedback, and have incorporated them into the revised version of the manuscript. We believe that these revisions have strengthened the quality and clarity of the research presented. We appreciate your thorough review and constructive feedback, which has greatly contributed to the improvement of the manuscript.

Reviewer 3 Report (New Reviewer)

The authors tried to review the inactivated COVID-19 vaccines in the China with the main focus is on vaccine safety and efficacy in special populations. The review is important and timely but I want to know why only special population?. whats the difference among vaccine for other population. 

I have few suggestion. 

mention more about the regulatory aspect.

there are alot of clinical trial ongoing and author just mention only 3 three references. add more.

add more tables regarding the description of different vaccines.

Author Response

Dear Editors and Reviewers:

Thank you for your efforts to review our manuscript entitled “A Review of Inactivated COVID-19 Vaccines Development in China: Focusing on Safety and Efficacy in Special Populations” (ID: vaccines-2401664). ”. We highly appreciate your time and comments that help a lot to improve the quality of the manuscript.

 We have addressed the comments from reviewers in a point-by-point way. Please see them in the response letter as below. In the revised manuscript, we have marked the revised texts in red color.

Hopefully, the resived manuscript will meet with the criteria of publication of your well-reputed journal. Thank you in advance.

Lidan hu and jianhua mao

On behalf of all the coauthors

  1. The authors tried to review the inactivated COVID-19 vaccines in the China with the main focus is on vaccine safety and efficacy in special populations. The review is important and timely but I want to know why only special population? whats the difference among vaccine for other population.

Response: Thank you for your feedback and interest in our manuscript. We appreciate your question regarding the focus on special populations in our review of inactivated COVID-19 vaccines in China. The reason for our emphasis on special populations is their unique susceptibility and potential challenges in vaccine response. Special populations, such as children, adolescents, older adults, cancer patients, and HIV-infected individuals, often have distinct immune characteristics and may require tailored vaccine strategies. By addressing the safety and efficacy of inactivated COVID-19 vaccines specifically in these populations, we aim to provide comprehensive insights that can guide healthcare professionals and policymakers in making informed decisions.

However, we acknowledge the importance of considering the broader population as well. It's important to note that our review does not exclude the general population, and the safety and efficacy data from clinical trials involving the general population are also considered in the overall assessment of inactivated COVID-19 vaccines. While the fundamental principles of vaccine development and evaluation remain the same for all population groups, it's important to note that the overall safety and effectiveness of these vaccines are similar across different population groups. Inactivated COVID-19 vaccines have undergone rigorous testing and have demonstrated satisfactory results in terms of generating immune responses and providing protection against the virus. In our revised manuscript, we also added these sentences in the conclusion part (page 12, line 488). Thank you again for your valuable input, which will undoubtedly strengthen the clarity and comprehensiveness of our manuscript.

  1. - have few suggestion. mention more about the regulatory aspect.

Response: Thank you for your valuable feedback. We appreciate your suggestion to provide more information about the regulatory aspect in our manuscript. In the revised version, we have expanded on the topic of regulatory approval (page 4, line 136).

  1. there are alot of clinical trial ongoing and author just mention only 3 three references. add more. add more tables regarding the description of different vaccines.

Response: Thank you for your valuable feedback. We appreciate your suggestion to include more references and provide additional tables describing different vaccines in our manuscript. We have expanded the number of references and table description in the manuscript to provide a more comprehensive overview of the ongoing clinical trials for COVID-19 vaccines (page 6, line 264).

Round 2

Reviewer 1 Report (Previous Reviewer 3)

I appreciate your last amendments and additional information especially in the Conclusions.

I emphasize the need for not including under efficacy criteria the logostics, service delivery and population acceptance. These are effectiveness criteria and more context specific .They are not the purpose of this publication. Effectiveness analysis in China would be complex .

Within the same line of thinking, i suggest  that you mention in the discussions "To reduce the infection and the disease impact,  all operational aspects of vaccination delivery have to be of course considered for the sake of effective vaccination performances and results. This study focuses on safety and efficacy"

Author Response

Thank you for your valuable feedback on our revised manuscript. We appreciate your recognition of the amendments . Regarding your suggestion on excluding logistics, service delivery, and population acceptance from the efficacy criteria, we agree that these aspects are more aligned with effectiveness analysis and are context-specific. We acknowledge that conducting effectiveness analysis in China would be complex and may go beyond the scope of our current publication, which focuses on the safety and efficacy of the vaccines. Thus,we have deleted these sentences in the revised manuscript. In line with your thinking, we will include a statement in the Discussions section to clarify that our study primarily focuses on the safety and efficacy of the vaccines (line 11, page 445). We appreciate your guidance in highlighting this distinction. We are grateful for your valuable suggestions, and we assure you that we will carefully consider and incorporate them into our manuscript. Thank you for your time and expertise in reviewing our work.

This manuscript is a resubmission of an earlier submission. The following is a list of the peer review reports and author responses from that submission.

Round 1

Reviewer 1 Report

Due to the evolving nature of SARS-CoV-2 variants the value of reviews of clinical trials on vaccines is limited and that is particularly the case for a review focused on inactivated viral vaccines.  Without a better understanding of the viral variants used to prepare the vaccines and comparative data there is not much to learn.   This review, often re-reporting the results of other meta-analyses and reviews, offers a number of opinions on the safety and efficacy of inactivated SAR-CoV-2 vaccines in various target populations often without appropriate citations and with little comment on the many other types of vaccines in widespread use.

Minor grammatical editing would improve the manuscript.

Reviewer 2 Report

The manuscript "A Review of Inactivated COVID-19 Vaccines: Focusing on Safety and Efficacy in Special Populations" by Hu et al., reviews the current development of inactivated COVID-19 Vaccines. The authors describe the progress of inactivated COVID-19 Vaccines development in China and discuss the application of inactivated COVID-19 vaccine for specific populations. As information is mainly gathered from China, I suggest changing the manuscript title and indicating clearly the discussion is focusing on China. For example, "A Review of Inactivated COVID-19 Vaccines Development in China: Focusing on Safety and Efficacy in Special Populations"

Part 2.2 - 2.7

- The authors could elaborate more on the details. For example, in part 2.4, the authors mention the important role of adjuvants in enhancing the immune response. Could the authors provide any examples? What is the mechanism? In part 2.5, what will be the biological indicators to justify the immunogenicity and safety level of the tested vaccine candidates?

Table 1 

- Authors could include more details about the sample size, clinical trial duration, immunogenicity, and safety data in each clinical trial.

- What are the criteria for ranking the clinical trial in this table? According to the starting time of the clinical trial? Status? 

- Are all clinical trials using the inactivated vaccine in their study? for example, NCT05043246/1, NCT04771559, NCT05043129

Reviewer 3 Report

The subject is so critical from a global public health point of view that the scientific based evidence of any statements should be carefully given.

A lot of issues needs to be clarified:

- Is this a global or a China specific review? Many other countries have reported safety and efficacy results . 

- References to the interim recommandations of the WHO Strategic Adisory Group of experts and to the WHO Prioritization Map have to be seriously considered

- Among the priority population groups, what about pregnant women, immunodepressed people, comorbidity factors, infected patients , 

- What about the duration of protection, the booster effects, the interchangeability with other vaccines, the variant mutation, the contraindications?

- what about the effectiveness criteria which include the logistics, the acceptabilty, the delivery facilities and competence?

- It is mentionned that the inactivated vaccines efficacy will reduce the infections and hospîtalizations? In principles yes, but any data based evidence?

- Usually, after the finding analyses, there is a section on "Discussions" . This implies the sterghts and weakness of the research/study and suggestions fot future research development.

In summary, this review has to be seriously consolidated